# Genotype-phenotype correlations of germline mutations in exon 10 of the *RET* proto-oncogene from 14 MEN2A families of Ethnic Han Chinese

Feng Li[1,2◉], Weiying Chen[2◉], Zhenyu Chen[2◉], Bijun Lian[1], Xudong Fang[1], Hangyang Jin[1], Huihong Wang[1], Jianqiang Zhao[3], Yiming Zhang[1]*, Xiaoping Qi[1,2]*

1 Department of Oncologic and Urologic Surgery, The 903rd PLA Hospital, Hangzhou Medical College, Hangzhou, Zhejiang Province, China, 2 Department of Urology, Taizhou Hospital of Zhejiang Province Affiliated to Wenzhou Medical University, Enze Hospital of Hangzhou Medical College, Taizhou Enze Medical Center (Group), Taizhou, Zhejiang Province, China, 3 Department of Head and Neck Surgery, Zhejiang Cancer Hospital, Hangzhou, Zhejiang Province, China

◉ These authors contributed equally to this work.
* qxplmd@163.com (QX); hz117zym@163.com (YZ)

## Abstract

### Objective

To investigate the genotype-phenotype correlations of multiple endocrine neoplasia type 2A (MEN2A) caused by mutations in exon 10 of the *RET* gene in Ethnic Han Chinese.

### Methods

A retrospective analysis was conducted on the family history and genetic characteristics of 14 independent MEN2A pedigrees, all carrying exon 10 mutations of the *RET* gene, from July 2003 to August 2023.

### Results

A total of 74 out of 133 participants carried germline mutations in exon 10 of the *RET* gene. The cohort included 26 males and 48 females, with nine types of mutations observed: p.C609R, p.C611F/Y, p.C618G/R/S/Y and p.C620R/S. Of these, the C618 mutation was the most prevalent (71.6%), followed by p.C611 (22.9%), p.C620 (4.1%), and p.C609 (1.4%). The penetrance rates for medullary thyroid carcinoma (MTC), pheochromocytoma, hyperparathyroidism, hirschsprung disease, and cutaneous lichen amyloidosis were 90.3%, 6.9%, 2.8%, 1.4% and 1.4%, respectively. Among the 72 patients with available clinical information, 41 (56.9%) exhibited symptoms of MTC. Comparison of the age at diagnosis, size of MTC, and the positive rate of cervical lymph node metastasis (N1) revealed significant differences between patients with symptomatic and asymptomatic MTC (all $P < 0.05$). There was

**Data availability statement:** All relevant data are within the paper and its Supporting Information files.

**Funding:** National Natural Science Foundation of China (81472861,XiPing Qi), the Key Project of Zhejiang Province Science and Technology Plan (2014C03048-1,XiPing Qi), Hangzhou Municipal Commission of Health and Family Planning Science and Technology Program (OO20190253,Feng Li; B20210355,Bijun Lian), and Taizhou Social Development Science and Technology Planning Project (23ywb56,Wei-ying Chen). The funders had no role in study design, data collection and analysis, decision to publish, or preparation of the manuscript.

**Competing interests:** The authors have declared that no competing interests exist.

a significant difference in the positivity rate of N1 between patients with the p.C618/C620 mutations and those with the p.C609/C611 mutations. Additionally, there was a significant difference in the initial serum calcitonin levels between N1 and N0 patients (P < 0.05).

## Conclusion

Exon 10 mutations of the *RET* gene are frequently located in codon 618 and contribute to the familial MTC phenotype. To improve the recognition of MEN2A, integrating family history, testing for *RET* mutations, and monitoring serum calcitonin levels are essential for early diagnosis and personalised treatment.

## Introduction

Multiple endocrine neoplasia type 2A (MEN2A) is a rare autosomal dominant cancer syndrome characterised by the presence of two or more specific endocrine tumours, including medullary thyroid carcinoma (MTC), pheochromocytoma (PHEO), and hyperparathyroidism (HPTH) [1–6]. MEN2A manifests in four variants [1]: ①classic MEN2A; ②MEN2A with cutaneous lichen amyloidosis (CLA); ③MEN2A with Hirschsprung disease (HSCR); ④familial MTC (FMTC). Mutations in the *RET* (rearranged during transfection) proto-oncogene, located on chromosome 10q11.2, are responsible for the development of MEN2A. Approximately 95% of MEN2A cases are attributed to mutations in exons 10 and 11 of the *RET* gene. Recently, the proportion of patients with exon 10 mutations has increased from 8% to 25%, with an increasing number of cases being reported [3–6]. However, most studies have focused on single families or individual cases with exon 10 mutations, and few studies have examined multiple families with this mutation [7–11]. In this study, we retrospectively analysed the clinical data of 74 MEN2A patients from 14 families to investigate the genotype-phenotype correlations in patients with MEN2A caused by mutations in exon 10 of the *RET* gene.

### Materials and methods

#### Participants

We selected 14 unrelated MEN2A families, each with exon 10 mutations in the *RET* gene, all of whom were diagnosed and treated at the 903rd PLA Hospital between 01/07/2003 and 31/08/2023. A comprehensive familial investigation was conducted, and clinical data, peripheral blood DNA specimens and serum samples were collected from 133 family members. For comparison, we used data from the International *RET* Exon 10 Consortium, which includes 340 patients from 103 families across 15 countries [10]. The required data were accessed on 15/11/2023 for research purposes, and all personal information was de-identified prior to analysis. MEN2A diagnostic criteria were based on the revised American Thyroid Association Guidelines for the management of MTC [1]. This study protocol was approved by the

Ethics Committee of the 903rd PLA Hospital, and informed consent was obtained from all participants or their family members. After data collection, the authors had no access to any information that could identify individual participants.

Imaging examinations and serum calcitonin levels (normal reference values: male <8.4 ng/L; female <5 ng/L) were measured for all 133 family members. The imaging examinations included thyroid, parathyroid, adrenal Doppler B-ultrasound, computed tomography, magnetic resonance imaging and, where clinically indicated, positron emission tomography. Additional biochemical tests were conducted, including those for carcinoembryonic antigen, parathyroid hormone (PTH), blood calcium, phosphorus, blood/urine catecholamines and plasma metanephrine (MN)/normetanephrine (NMN). These tests were performed on patients with *RET* mutations, as described in previous studies [3,8,9].

Following the confirmation of *RET* mutations and elevated levels of Ctn or MNs, prophylactic thyroidectomy or surgical treatment was performed to diagnose MTC or PHEO. Tumour staging for MTC was performed according to the American Joint Committee on Cancer version 7 tumour-node-metastasis (TNM) classification system [12]. Individuals were followed up on until August 2023.

## Detection of *RET* mutation

Peripheral blood samples (5 mL) were collected from all 133 family members, including the probands. Targeted sequencing was performed using an Illumina HiSeq 2000 Analyser. The sequencing results were validated using Sanger sequencing with an ABI 3700 Genetic Analyser (PerkinElmer, Fremont, CA) [3,8,9].

## Statistical analysis

IBM SPSS 22.0 software was used for data processing and analysis. Normally distributed quantitative data were expressed as mean ± standard deviation, and between-group comparisons were performed using a *t*-test. Categorical data were expressed as constituent ratios (%), and between-group comparisons were performed using the $\chi^2$ test or Fisher's exact test. The Kaplan-Meier method was used to plot the age-related penetrance curve of MTC, and the Log-rank test was used to compare differences between the age-related MTC penetrance curves of patients with different codon mutations of *RET*. A significance level of $\alpha = 0.05$ was used, and $P < 0.05$ was considered statistically significant.

## Results

### Clinical features and phenotypic data

Among the 133 participants, 74 carried germline *RET* mutations and were diagnosed with MEN2A, including 26 males and 48 females (Table 1). Of these 74 patients, two males with the p.C618R mutation were excluded due to insufficient clinical information, leaving 72 patients for analysis (S1 Table). The mean age at diagnosis for the remaining 72 patients was 39.0 ± 18.1, with a range from 4 to 83 years. Thirty-nine patients (54.2%) presented with symptoms of MTC, including palpable thyroid nodules, lymph node enlargement, neck pain, and/or MTC-related diarrhoea. The remaining 33 patients (45.8%) were diagnosed through family screening and *RET* mutation analysis. A significant difference was observed in the age at diagnosis between the symptomatic and asymptomatic groups: 50.9 ± 18.1 (21–83) years in the symptomatic group vs 32.6 ± 18.8 (4–69) years in the asymptomatic group (P<0.001). In the asymptomatic group: ① six patients (8.3%) showed no evidence of MTC, as no thyroid nodules were found by B-ultrasound, and their Ctn levels were normal. The mean age at diagnosis for this group was 21.2 ± 16.5 (5.2–53) years, including four children with a mean age of 10.5 ± 3.2 (5.2–13.4) years and two adult males aged 32 and 53 years; ② one patient (1.4%) had thyroid C-cell hyperplasia (CCH), a 26-year-old female with a Ctn level of 5.78 ng/L but no obvious thyroid nodules detected by ultrasound; ③ twenty-four patients (33.3%) had no MTC symptoms (i.e., no palpable thyroid nodules, but had imaging and serological evidence of MTC), with a mean age at diagnosis of 35.2 ± 18.8 (4–74) years. Additionally, two female patients (2.8%) presented with MTC symptoms and were diagnosed through family investigation, aged 37 and 41 years. In addition, among the 72

Table 1. Clinical characteristics of 14 MEN2A pedigrees with *RET* mutations located in exon 10.

| Codon | Nucleotide | RET mutation | Number of pedigrees (%) | RET mutation carriers (cases; %) | Gender (Male/Female) | Number of cases (%) | | | | Phenotype (number of families) | number of cases (age range: years) | | | ADM (years) | ADP (years) |
| --- | --- | --- | --- | --- | --- | --- | --- | --- | --- | --- | --- | --- | --- | --- | --- |
| | | | | | | MTC | PHEO | HPTH | HSCR | | HPTH | CLA | HSCR | N1M0 | M1 |
| 609 | c.1825T>C | p.C609R | 1 (7.1) | 1 (1.4) | 0/1 | 1 | 0 | 0 | 0 | FMTC(1) | | 1 (52) | – | 52 | – |
| 611 | c.1832G>A | p.C611Y | 1 (7.1) | 16 (21.6) | 8/8 | 14 | 2 | 0 | 0 | MEN2A-CLA(1) | | 7 (30-61) | – | 42.8 (20-74) | 53 (37,69) |
| | c.1832–1833delGCinsTT | p.C611F | 1 (7.1) | 1 (1.4) | 1/0 | 0 | 1 | 1 | 0 | MEN2A(1) | | – | – | – | 32 |
| 618 | c.1852T>G | p.C618G | 2 (14.3) | 4 (5.4) | 2/2 | 4 | 0 | 0 | 0 | FMTC(2) | | 2 (37,44) | – | 57.8 (37-83) | – |
| | c.1852T>C | p.C618R | 4 (28.6) | 27* (36.5) | 8/19 | 24 | 2 | 1 | 1 | MEN2A(3)/MEN2A-HSCR(1) | | 12 (21-59) | 1 (45) | 37.5 (9-65) | 50 (41,59) |
| | c.1852T>A | p.C618S | 1 (7.1) | 13 (17.6) | 3/10 | 12 | 0 | 0 | 0 | FMTC(1) | | 7 (29-64) | – | 34.9 (4-64) | – |
| | c.1853G>A | p.C618Y | 1 (7.1) | 9 (12.2) | 2/7 | 8# | 0 | 0 | 0 | FMTC(1) | | 5 (24-59) | 1 (60) | 49 (24-78) | – |
| 620 | c.1858T>C | p.C620R | 1 (7.1) | 1 (1.4) | 1/0 | 1 | 0 | 0 | 0 | FMTC(1) | | 1 (27) | – | 27 | – |
| | c.1859G>C | p.C620S | 2 (14.3) | 2 (2.7) | 1/1 | 2 | 0 | 0 | 0 | FMTC(2) | | 2 (25,39) | – | 32 (25,39) | – |
| Total | | | 14 (100) | 74* (100) | 26/48 | 65 (90.3) | 5 (6.9) | 2 (2.8) | 1 (1.4) | FMTC(8) MEN2A(4) | | 37 (21-64) | 2 (45-60) | 40.9 (4-83) | 47.6 (32-69) |

MTC, medullary thyroid carcinoma; PHEO, pheochromocytoma; HPTH, hyperparathyroidism; CLA, cutaneous lichen amyloidosis; HSCR, Hirschsprung disease; N1M0, cervical lymph node metastasis of MTC, only; M1, distant metastasis of MTC; ADM, diagnostic age of MTC; ADP, diagnostic age of PHEO; FMTC, Familial MTC; MEN2A, classic MEN2A; #includes 1 case of a p.C618Y patient with C-cell hyperplasia (CCH); *includes 2 male p.C618R patients from different pedigrees, lacking initial clinical data.

patients with MEN2A, 5 (6.9%) had unilateral PHEO, 2 (2.8%) had HPTH, and 1 patient (1.4%) each had CLA and HSCR. The penetrance of MTC in the Asian population was significantly higher than that in the non-Asian population (90.3% vs. 77.4%, P = 0.013), while the penetrance of PHEO was lower in the Asian population than that in the non-Asian population (6.9% vs. 16.9%, P = 0.033), as compared to the data in reference 10. However, no significant difference was observed in the penetrance of HPTH and HSCR (Fig 1).

### Identification of the *RET* germline mutations

Of the 74 patients, nine types of *RET* germline nucleotide changes in exon 10 of the *RET* gene were identified, including eight point mutations: p.C609R (c.1825T > C), p.C611Y (c.1832G > A), p.C618G/R/S/Y (c.1852T > G, c.1852T > C, c.1852T > A, and c.1853G > A), p.C620R/S (c.1858T > C and c.1859G > C), and one deletion-insertion mutation: p.C611F (c.1832–1833delGCinsTT). The C618G/R/S/Y mutation at codon 618 was the most common, found in 53 patients (71.6%), followed by the C611F/Y mutation at codon 611 in 17 patients (22.9%), the C620R/S mutation at codon 620 in 3 patients (4.1%), and the C609R mutation at codon 609 in 1 patient (1.4%). In terms of familial distribution, the C618 mutation was observed in 8 families (57.1%), the C620 mutation in 3 families (21.4%), the C611 mutation in 2 families (14.3%), and the C609 mutation in 1 family (7.1%), as shown in Table 1. When compared with the non-Asian populations in reference 10, no statistically significant difference was found in the distribution of various mutation types (P = 0.588) (Fig 2).

### Relationship between phenotype and *RET* genotype

The patients belonged to four different MEN2A variants: ①Classic MEN2A: Sixteen patients were from 4 families (28.6%), including 1 patient from a C611F family (PHEO/HPTH) and 15 patients from 3 C618R families. ②MEN2A with CLA: Sixteen patients belonged to 1 C611Y family (7.1%). ③MEN2A with HSCR: Ten patients were from 1 C618R family (7.1%). ④FMTC: Thirty patients were from 8 families (57.1%), including 1 patient from a C609R family, 13 patients from 1 C618G family, 9 patients from 1 C618Y family, 1 patient from a C620R family, 4 patients from 2 C618S families, and 2 patients from 2 C620S families. Further details are provided below.

### MEN2A-MTC characteristics

Among the 72 patients, 66 presented with MTC/CCH, and 6 had no evidence of MTC. The incidence rate of MTC was 90.3% (65/72), with a mean age at diagnosis of 40.8 ± 17.7 (4–83) years and a mean tumour size of 2.0 ± 1.1 (0.4–4.5) cm. The presence of exon 10 mutations in *RET* was strongly associated with the development of MTC, with the following age-related penetrance: 3.2% by age 11 years, 6.5% by age 20, 32% by age 30, 47% by age 37, 84% by age 60, 95% by age 70, and 100% by age 83 (Fig 3).

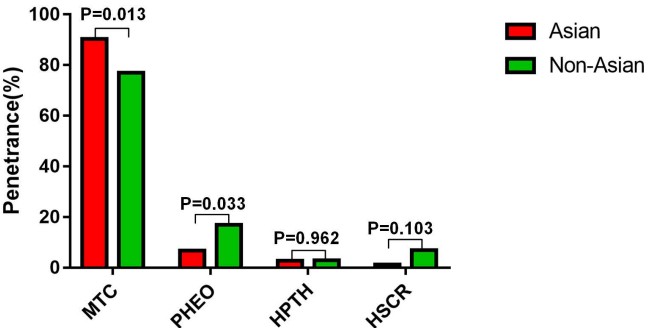

**Fig 1. Penetrance of MTC, PHEO, HPTH and HSCR between Asian and non-Asian populations.**

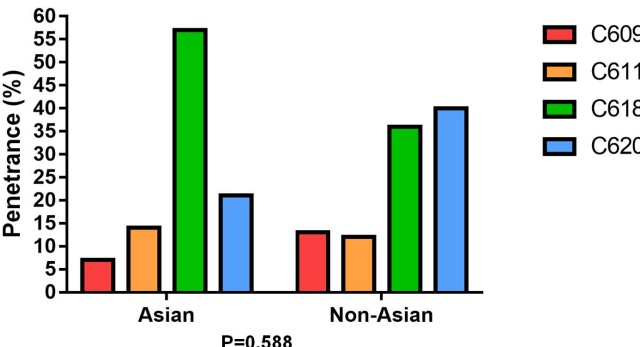

**Fig 2. Distribution of various mutation types between Asian and non-Asian pedigrees.**

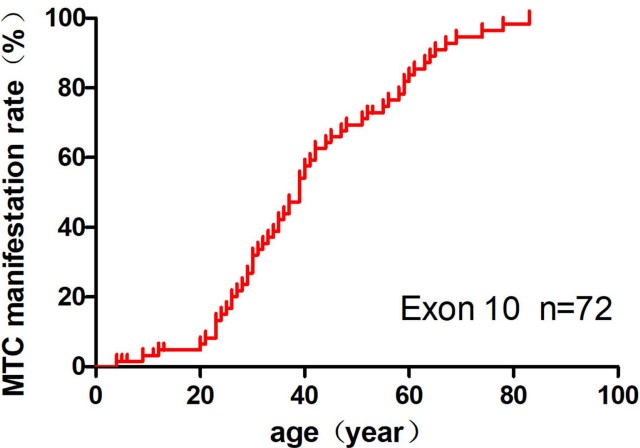

**Fig 3. Age-related penetrance of MTC with *RET* mutations in exon 10.**

When, comparing 41 patients with symptomatic MTC to 24 patients with asymptomatic MTC, statistically significant differences were found in the mean age at diagnosis of MTC [43.1±14.4 (21–83) years vs. 32.8±18.6 (4.6–69) years], MTC size [2.4±0.8 (1.2–4.5) cm vs. 0.8±0.3 (0.4–1.4) cm], and positive rate of cervical N1 [100% (31/31) vs. 35.3% (6/17)] ($P=0.012$, $P=0.000$, $P=0.000$). Comparison of the mean age at diagnosis (32.3±18.7 years vs. 35.0±19.7 years) and age-related MTC penetrance rates between patients with C618/C620 and C609/C611 mutations showed no significant differences ($P=0.621$, $P=0.722$; Fig 4). However, statistically significant differences were found in the positive rate of cervical lymph node metastasis of MTC [N1:90.9% (29/33) vs. 60% (8/15)], confirmed by pathological examination and MTC size (2.2±0.9 cm vs. 1.3±1.2 cm) ($P=0.085$, $P=0.027$).

A comparison of the initial Ctn levels between patients with cervical negative (N0; 11 cases) and positive (N1; 37 cases) lymph nodes, confirmed by pathological examination, revealed a statistically significant difference (31.3±19.7 (9.9–67.1) ng/L vs. 1383.9±730.1 (17.6–2652) ng/L, $P=0.000$).

In addition, 2 out of 72 patients had distant metastasis. One patient with the C618R mutation had liver metastasis, and the other with the C618Y mutation had bone and lung metastasis, both with Ctn levels >2000 ng/L. Of the 8 children diagnosed with MEN2A before the age of 14, the mean age at diagnosis was 9.4±3.3 (4.6–13.4) years. No thyroid nodules were detected on B-ultrasound examination, and serum Ctn levels ranged from 2.94 to 12.2 ng/L. Four of these children,

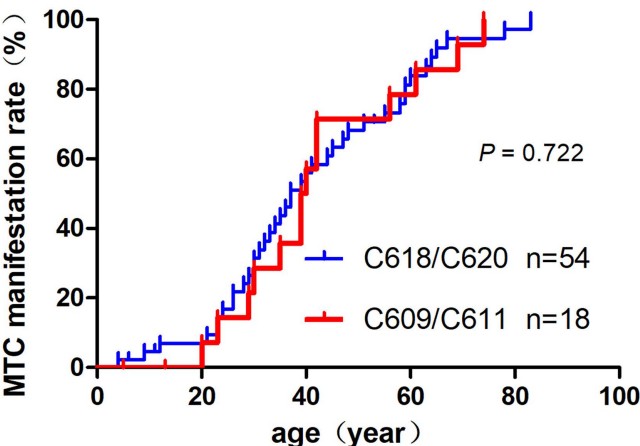

**Fig 4. Comparison of age-related penetrance of MTC in exon 10 *RET* mutations.**

including the youngest patient at 4.6 years old, had slightly elevated Ctn levels (1 with the C618R mutation and 3 with C618S), while the remaining 4 children had normal Ctn levels.

## MEN2A-PHEO characteristics

Among the 5 (6.9%) patients presenting with PHEO, the mean age at diagnosis was 47.6±14.1 (32–69) years, and the tumour size ranged from 2.6 cm to 9.5 cm in the longest diameter. Three of the patients were females, and two were males; all had unilateral PHEO. Three patients had left adrenal PHEO, and two had right adrenal PHEO. The mutations identified were C611 in 3 patients (17.6%) and C618 in 2 patients (3.9%). Among these patients, 2 were diagnosed due to high blood pressure, 2 were diagnosed through family investigation, and 1 was incidentally found during liver and gall-bladder imaging. Four of these patients were associated with MTC (1 had PHEO 3 years prior to MTC, 1 was diagnosed simultaneously with PHEO and MTC, and 2 had PHEO 1 and 6 years after MTC surgery, respectively). One patient had no clinical or biochemical evidence of MTC.

## Other MEN2-associated diseases

Two of the 72 patients (2.8%) had HPTH, one with the C611Y mutation and one with the C618R mutation. The age at diagnosis was 39 or 23 years, respectively. Additionally, one patient (1.4%) with CLA harbouring the C611Y mutation was diagnosed at 40 years of age. This patient had experienced itching and scratching in the scapular region of the upper back for 18 years. One patient (1.4%) with HSCR carrying the C618R mutation was diagnosed at 15 years of age, which was earlier than the age of diagnosis for MTC, which occurred at 33 years (Table 1).

## Outcome of treatment

Of the 72 patients, six opted for watchful waiting or strict follow-up monitoring owing to a lack of evidence of MTC, and 17 refused thyroid surgery due to concerns about surgical risks and its potential impact on quality of life. The remaining 49 patients (68.1%) underwent bilateral total thyroidectomy (TT). Among these, one patient underwent TT alone, 33 underwent TT with bilateral level VI lymph node dissection, and 15 underwent TT with level II-VI lymph node dissection. Postoperatively, 48 of the 49 patients with MTC were confirmed histopathologically, while the remaining patient with the C618Y mutation was diagnosed with CCH. All patients with the C609R, C618G/S/Y and C618S/R mutations had positive lymph nodes. The positive rates of lymph nodes involvement in patients with the C611Y and C618R mutations were 7/14

(50%) and 12/16 (75%), respectively. Furthermore, of the five patients (6.9%) with concomitant PHEO, four underwent adrenal-sparing laparoscopic adrenalectomy, while one underwent laparoscopic total adrenalectomy (PHEO size, 9.5 cm). After a mean follow-up of 88.6±68.5 (7–192) months, no ipsilateral recurrence or contralateral PHEO development was observed.

## Discussion

MEN2A, also known as Sipple syndrome, is primarily caused by germline mutations in the *RET* gene in nearly all cases. Exceptions include two MTC families with germline mutations in the *ESR2* gene (c.948delT) and *MET* gene (c.1250G>A) [1–6]. Mutation hotspots are primarily concentrated in exons 8, 10, 11, and 13–15 of the *RET* gene. In recent years, with the identification of moderate-risk mutations at codons 609, 611, 618, and 620 in exon 10 among MEN2A cases, the relative prevalence of high-risk C634 mutations has decreased from 87% to 40%. In this series of 14 MEN2A pedigrees, all 74 patients had mutations in *RET* involving all four codons in exon 10, with C618 (57.1%), C620 (21.4%), C611 (14.3%), and C609 (7.1%) mutations. The C618 mutations had the highest incidence of 71.6%, while the C609 mutations were the least common (1.4%). These findings suggest that there may be familial and/or regional aggregation of *RET* mutations [10,11].

Mutations in *RET* exons 10 and 11 result in the substitution of cysteine residues within the extracellular domain. These substitutions lead to the formation of aberrant homodimers through disulfide bond-mediated self-aggregation, which subsequently induces the phosphorylation of intracellular tyrosine residues. This process activates downstream signal transduction pathways, triggering a cascade of reactions that promote excessive cell proliferation and contribute to the pathogenesis of MEN2A [1]. Machens revealed that the C630/C634 residues in exon 11 are 2–6 amino acid residues away from the cell membrane, compared to the C620/C618/C611/C609 residues in exon 10, which are 16/18/25/27 amino acid residues away from the cell membrane, respectively. *RET* mutants closer to the cell membrane exhibit a higher incidence of MTC progression and PHEO penetrance [11]. In this study, MTC penetrance was approximately 50% by the age of 37 years, and the mean age at diagnosis for patients with clinical symptoms of MTC was approximately 43.1 years. This finding is similar to previous reports where the mean age at thyroid surgery was around 37 years, in contrast to patients with the C630/C634 mutation in exon 11, who had a mean age of only 22.9 years ($P < 0.001$) [11]. The penetrance rate of PHEO in this study was 6.9%, which was significantly lower than that of 52%–67.1% observed in patients with the C634 mutation, who were near 50 years of age [1,4]. In addition, our findings showed that, although no significant difference in the age at diagnosis and MTC penetrance was observed between patients with C618/C620 mutations and those with C609/C611 mutations, patients with C618/C620 mutations had a higher risk of cervical lymph node metastasis. This suggests that in clinical practice, more comprehensive examinations are required for such patients. [1,5,11]. The majority of MEN2A-CLA cases are associated with C634 (98.2%) or V804M mutations (1.8%). However, in this series, two patients with the C611Y mutation exhibited clinical features of (MTC+PHEO+CLA) or (MTC+ pruritus in the scapular region of the upper back), indicating a rare and diverse genotype-phenotype relationship in MEN2A-CLA [4,7]. When comparing the tumour-specific penetrance rates of 340 MEN2A patients with exon 10 *RET* mutations, as reported in reference 10, the penetrance of MTC in the Asian population was significantly higher than that in the non-Asian population, while the penetrance of PHEO was lower in the Asian population. It can be inferred that the differing penetrance rates of MTC and PHEO between the two populations were due to the later age of MTC diagnosis and earlier age of PHEO diagnosis in this study [10]. The penetrance of MEN2A-MTC/PHEO may also be influenced by factors such as race, regional distribution, and sample size. Additionally, research suggests that the same type of *RET* mutation can lead to different clinical phenotypes and disease progression [1,5,13]. Further studies are required to elucidate the specific molecular mechanisms underlying this process.

Early diagnosis and standardised treatment are essential for improving tumour-free survival rates and reducing mortality in patients with MEN2A [1,2,4,13]. In the present study, pedigree investigation and *RET* mutation screening were conducted, revealing that 43.1% of all MEN2A patients were asymptomatic. The mean age at diagnosis, MTC long

diameter, and positivity rate for cervical N1 were significantly lower than those in patients with symptomatic MTC (all $P < 0.01$). Pedigree investigation and *RET* mutation screening are useful tools for the early diagnosis of MEN2A [1,2,4,13]. Genotype-phenotype analysis indicated that patients with the C618/C620 mutation had a significantly higher positive rate of cervical N1 than those with the C609/C611 mutation ($P = 0.011$), suggesting that both TNM staging and *RET* mutation genotypes should be considered in the clinical diagnosis and treatment of MEN2A-MTC for comprehensive evaluation and analysis [1,2,4,13]. In addition, we found that Ctn levels ranged from 9.87 to 67.1 ng/L before initial surgery in cases where MTC was limited to the thyroid gland (N0), indicating that patients who underwent a bilateral TT without cervical lymph node dissection could achieve a cure. Conversely, when levels of Ctn exceeded 100 ng/L, all patients had cervical N1, and bilateral TT with cervical level VI lymph node dissection was necessary. If Ctn levels are within normal limits, watchful waiting or strict follow-up monitoring is recommended to delay surgery without compromising treatment efficacy and to reduce or avoid complications such as recurrent laryngeal nerve injury and hypoparathyroidism [1,2,4,13]. Preoperative integration of Ctn levels and *RET* mutation testing is essential for the precise selection of the timing and extent of surgery for MEN2A-MTC [1–4,13–15]. Unfortunately, four children with slightly elevated Ctn levels, whose parents refused surgery, underwent early bilateral TT. In contrast, the other six patients, who had no evidence of MTC (two of whom were already 32 and 53 years old), should be closely monitored. Bilateral TT should not be performed until serum or stimulated Ctn levels are elevated [1–4,13]. For two patients with MTC and liver, bone, and lung metastases along with elevated Ctn (>2000 ng/L), comprehensive treatment, including high-selectivity *RET* inhibitors, such as pralsetinib or selpercatinib, is recommended [4,16,17].

The age of onset of MEN2A-related PHEO associated with *RET* mutation in exon 10 is relatively late (median age ~40 years), with a low prevalence (~17%) and bilateral incidence rates (3.6%–4.7%) [5,11]. In the present study, 5 patients (6.9%) presented with unilateral PHEO, with a mean age at diagnosis of 47.6 ± 14.1 (32–69) years, involving 2 codons in exon 10: C611 (17.6%) and C618R (3.9%) (Table 1). This shows a relatively single and slightly different form, which may be related to sample size and the bias of *RET* phenotypes. Of the five patients with PHEO, all were adequately treated with terazosin hydrochloride or phenoxybenzamine and volume expansion for 1–2 weeks. Four patients underwent adrenal-sparing adrenalectomy, whereas the remaining patient underwent total adrenalectomy due to a PHEO size of 9.5 cm. Due to the low risk of recurrence and reduced permanent adrenal cortical function, adrenal-sparing adrenalectomy (laparoscopy) is the preferred treatment for MEN2A-PHEO. However, if the tumour is large (>5 cm) or malignant (<1%), total adrenalectomy becomes necessary [4,18–20]. As this was a retrospective study with a relatively small sample size, there was some degree of selection and information bias. Future multicentre studies are needed to enhance the generalisability of these results.

## Conclusion

MEN2A, caused by *RET* mutations in exon 10, presents with unique clinical features. Increasing awareness of MEN2A, along with integrating family investigations, *RET* mutations testing, and Ctn levels measurements, is beneficial for early diagnosis and standardised treatment of MEN2A associated with exon 10 mutations. Focusing on molecular diagnosis and individualised management through the "5P diagnostic and therapeutic strategy" can improve the prognosis and quality of life of patients with MEN2A [1,2,4,21–23].

## Supporting information

**S1 Table. Clinical data of the 74 patients.**
(XLSX)

## Acknowledgments

We would like to thank Editage (www.editage.cn) for English language editing.

## Author contributions

**Conceptualization:** Feng Li, Xiaoping Qi.

**Data curation:** Feng Li, Bijun Lian, Hangyang Jin, Huihong Wang, Jianqiang Zhao, Xiaoping Qi.

**Formal analysis:** Feng Li, Zhenyu Chen, Bijun Lian, Xiaoping Qi.

**Funding acquisition:** Feng Li, Weiying Chen, Xiaoping Qi.

**Writing – original draft:** Feng Li, Xudong Fang, Xiaoping Qi.

**Writing – review & editing:** Yiming Zhang, Xiaoping Qi.

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
