## [Decision Letter · Decision Letter 0]

8 Apr 2025

Dear Dr. Qi,

Thank you for submitting your manuscript to PLOS ONE. After careful consideration, we feel that it has merit but does not fully meet PLOS ONE’s publication criteria as it currently stands. Therefore, we invite you to submit a revised version of the manuscript that addresses the points raised during the review process.

Your manuscrupt was reviewed by two experts in the field. Both identified many problems in your submission. Please review the attached comments and provide point-by-point responses.

We look forward to receiving your revised manuscript.

Kind regards,

Yury E Khudyakov, PhD

Academic Editor

PLOS ONE

“National Natural Science Foundation of China (81472861,XiPing Qi), the Key Project of Zhejiang Province Science and Technology Plan (2014C03048-1,XiPing Qi), Hangzhou Municipal Commission of Health and Family Planning Science and Technology Program (OO20190253,Feng Li; B20210355,Bijun Lian), and Taizhou Social Development Science and Technology Planning Project (23ywb56,Weiying Chen).”

Reviewers' comments:

Reviewer's Responses to Questions

**Comments to the Author**

1. Is the manuscript technically sound, and do the data support the conclusions?

Reviewer #1: Yes

Reviewer #2: Yes

2. Has the statistical analysis been performed appropriately and rigorously?

Reviewer #1: Yes

Reviewer #2: Yes

3. Have the authors made all data underlying the findings in their manuscript fully available?

Reviewer #1: Yes

Reviewer #2: Yes

4. Is the manuscript presented in an intelligible fashion and written in standard English?

Reviewer #1: Yes

Reviewer #2: Yes

Reviewer #1: 1-Enhance the discussion by including more comparisons with non-Asian populations and exploring the molecular mechanisms underlying phenotypic differences.

2-Incorporate more explanatory graphs to highlight differences between genetic subgroups, making the results clearer for readers.

3-Detail potential biases of the retrospective study, addressing limitations in the generalization of findings to broader populations.

4-Expand clinical applicability by providing recommendations for medical practice based on RET genotype, facilitating personalized management strategies.

Reviewer #2: Overall, I believe that the article makes an important contribution to our understanding of RET mutation clinical outcomes. I do not have much comments on various analyses presented in the article - this line on inquiry appears to be solid and well-thought. However, in the article aiming to investigate genotype-phenotype correlations I would expect to see much more discussion regarding the connections between different observed mutations and various clinical symptoms in the patients. While the authors make comparisons between the patients based on various observed symptoms and clinical outcomes, the vast majority of these analyses do not try to contemplate if specific genetic components play any significant role in the observed differences.

On a grammatical level, I noticed several sentences throughout the text with phrasing that felt awkward, if not outright incorrect. Additionally, the authors should consider breaking up long, multi-line sentences into smaller pieces to improve clarity. Overall, I believe the text would benefit from a good line-editing service.

Aside from the above, I am listing several point comments that also mainly revolve around typos and grammar:

- When listing corresponding authors there is no need to reiterate their affiliations, only state preferred methods of communication: mail, phone, etc.

- The phrasing “RET mutations within exon 10” throughout the text sounds like “RET” is the name of these specific mutations and not of the gene where they occur. Re-writing it as “exon 10 mutations of RET gene” makes the message more clear.

- Lack of line and page numbering makes it more cumbersome to make comments on particular parts of the text.

- Reference numbers throughout the text shouldn’t be in blue color and uppercase.

- On the title page the authors say that the text contains a single figure and a single table, but attach two figures to the text.

- Figures should contain legends, did not find them anywhere.

- Abstract:

- In the sentence “Comparison of the age at diagnosis...” change “there were” to “revealed” for correct grammatical structure.

- The sentence “Patients were divided into...” is not grammatically correct, please rewrite for clarity. Also, it is not clear from the sentence if N1 and N0 patients were compared between groups or inside each group of mutation carriers. And font size of “were divided into” for some reason is different from the rest.

- Introduction:

- The sentence “Recently, it has been proved...” is not entirely clear to me. Do the authors mean that improve in diagnostics lead to a higher number of identified cases? Or that overall amount of people carrying these mutations keeps increasing for some reason?

- In the sentence “most studies in the literatures describe only through single family”: 1) I think in the case here using “literature” is more correct and common; 2) “a” should be used instead of through”

- Materials and Methods:

- “exon10” - missing space.

- Results:

- The sentence: “In all, 74 of 133 participants who were found...” does not sound grammatically correct.

- Consider breaking long multi-line sentences into shorter segments for better clarity of the text.

- “pathologicaal” - extra “a” typo.

- References:

- Reference numbers should have same font color.

- Adding “[J]” after journal title is not a part of a reference style used by PLOS

- The volume number should be separated from the year of publication by a semicolon, not a comma.

- The references should include six author names before using et al., not three.

**Do you want your identity to be public for this peer review?** For information about this choice, including consent withdrawal, please see our Privacy Policy

Reviewer #1: No

Reviewer #2: No

---

## [Author Response · Author response to Decision Letter 1]

23 Jun 2025

Cover Letter for revised MS [ID: PONE-D-24-51352]

To: Editor, PLoS One

Reply to Referees' Comments on MS ID PONE-D-24-51352

Dear Editor and Reviewers,

We are very grateful for your constructive and thoughtful comments on our manuscript “Genotype-phenotype correlations of germline mutations in exon 10 of RET proto-oncogene from 14 MEN2A families in Ethnic Han Chinese” (PONE-D-24-51352). Those comments are all valuable and very helpful for revising and improving our paper. We have studied comments carefully and have made correction which we hope meet with approval.

Response to ediror,

Q1. Please ensure that your manuscript meets PLoS One's style requirements, including those for file naming.

A1. We have corrected the manuscript’ style according to PLoS One style templates.

Q2. Please state what role the funders took in the study. If the funders had no role, please state: "The funders had no role in study design, data collection and analysis, decision to publish, or preparation of the manuscript." If this statement is not correct you must amend it as needed. Please include this amended Role of Funder statement in your cover letter; we will change the online submission form on your behalf.

A2. We confirm that the funders had no role in study design, data collection and analysis, decision to publish, or preparation of the manuscript.

Q3. PLOS requires an ORCID iD for the corresponding author in Editorial Manager on papers submitted after December 6th, 2016. Please ensure that you have an ORCID iD and that it is validated in Editorial Manager. To do this, go to‘Update my Information’(in the upper left-hand corner of the main menu), and click on the Fetch/Validate link next to the ORCID field. This will take you to the ORCID site and allow you to create a new iD or authenticate a pre-existing iD in Editorial Manager.

A3. We confirm that the corresponding author already has an ORCID iD, and it has been successfully validated in Editorial Manager.

Q4. Your ethics statement should only appear in the Methods section of your manuscript. If your ethics statement is written in any section besides the Methods, please delete it from any other section.

A4. We have deleted any duplicate or misplaced ethics - related content from other sections to ensure compliance with your requirements.

Q5. Please include a separate caption for each figure in your manuscript.

A5. Thank you for your feedback! We have now added a separate caption for each figure in the manuscript, ensuring that every figure has a clear and accurate description.

Response to Reviewer 1,

Q1. Enhance the discussion by including more comparisons with non-Asian populations and exploring the molecular mechanisms underlying phenotypic differences.

A1. We have incorporated more comparisons and expounded on the possible reasons in our discussion. Compared the tumor-specific penetrance rate of 340 MEN2A patients with exon 10 of RET mutations reported in reference 10, the penetrance of MTC in the Asian population is significantly higher than that in the non-Asian population, while the penetrance of PHEO is lower than that in the non-Asian population. It can be inferred that the different penetrance rate of MTC and PHEO between the two populations is caused by the later diagnosis age of MTC and the earlier diagnosis age of PHEO in this study[10]. And the penetrance of MEN2A- MTC/PHEO may be also influenced by differences in race and regional distribution, as well as sample size. It may be also suggested that the same type of RET mutation can lead to different clinical phenotypes and disease progression [1,5,13]. The specific molecular mechanism may require further research to elucidate.”

Q2. Incorporate more explanatory graphs to highlight differences between genetic subgroups, making the results clearer for readers.

A2. We have created additional explanatory graphs to illustrate the differences between genetic subgroups, such as Fig 1and Fig 2. By incorporating these graphs, we hope to enhance the clarity of our results and make it easier for readers to interpret the differences.

Q3. Detail potential biases of the retrospective study, addressing limitations in the generalization of findings to broader populations.

A3. We have added the sentence“Since this is a retrospective study and the sample size is relatively small, there is a certain degree of bias, such as selection bias and information bias, in the study. It is hoped that multi-center studies can be carried out in the future to increase the generalizability of the results.”in the discussion section to address limitations in the generalization of findings to broader populations.

Q4. Expand clinical applicability by providing recommendations for medical practice based on RET genotype, facilitating personalized management strategies.

A4. We discussed the potential benefits of personalized management strategies based on RET genotype in the Discussion section from line from 299 to 306: “In addition, our study findings also showed that although there is no significant difference in the age at diagnosis and the penetrance of MTC between patients with C618/C620 mutations and those with C609/C611 mutations, patients with C618/C620 mutations have a higher risk of cervical lymph node metastasis which suggests that in clinical practice, more comprehensive examinations are required for such patients.”

By doing so, we hope to expand the clinical applicability of our study and contribute to the development of more personalized and effective medical practices.

Response to Reviewer 2,

Q1. On a grammatical level, I noticed several sentences throughout the text with phrasing that felt awkward, if not outright incorrect. Additionally, the authors should consider breaking up long, multi-line sentences into smaller pieces to improve clarity. Overall, I believe the text would benefit from a good line-editing service.

A1. We fully acknowledge the issues you pointed out. Regarding the awkward phrasings and potentially incorrect sentences, we have conducted a comprehensive grammar check. We also sought the help from native English speakers to identify and rectify these problems, ensuring the language used is both accurate and natural.

Q2. When listing corresponding authors there is no need to reiterate their affiliations, only state preferred methods of communication: mail, phone, etc.

A2. The corresponding authors’affiliations have been deleted.

Q3. The phrasing “RET mutations within exon 10” throughout the text sounds like “RET” is the name of these specific mutations and not of the gene where they occur. Re-writing it as “exon 10 mutations of RET gene” makes the message more clear.

A3. All the“RET mutations within exon 10”has been replaced to“exon 10 mutations of RET gene”.

Q4. Lack of line and page numbering makes it more cumbersome to make comments on particular parts of the text.

A4. The line and page number have been added to the manuscript.

Q5. Reference numbers throughout the text shouldn’t be in blue color and uppercase.

A5. All the reference numbers have been cancelled blue color and uppercase.

Q6. On the title page the authors say that the text contains a single figure and a single table, but attach two figures to the text.

A6. This illustration is not necessary, we have deleted it.

Q7. Figures should contain legends, did not find them anywhere.

A7. We have added figure legends in the last page.

-Abstract:

Q8. In the sentence “Comparison of the age at diagnosis...” change “there were” to “revealed” for correct grammatical structure.

A8. We have changed“there were”to“revealed”.

Q9. The sentence “Patients were divided into...” is not grammatically correct, please rewrite for clarity. Also, it is not clear from the sentence if N1 and N0 patients were compared between groups or inside each group of mutation carriers. And font size of “were divided into” for some reason is different from the rest.

A9. We have rewrote the sentence to“There was a significant difference in the positive rate of N1 between patients with p.C618/C620 mutations and those with p.C609/C611 mutations. Additionally, there was also a significant difference in the initial serum calcitonin levels between N1 and N0 patients (P < 0.05).”

- Introduction:

Q10. The sentence “Recently, it has been proved...” is not entirely clear to me. Do the authors mean that improve in diagnostics lead to a higher number of identified cases? Or that overall amount of people carrying these mutations keeps increasing for some reason?

A10. We have changed it to“Recently, it has been reported that the proportion of patients with exon 10 mutations has increased from 8% to 25% with more and more cases disclosed.”

Q11. In the sentence “most studies in the literatures describe only through single family”: 1) I think in the case here using “literature” is more correct and common; 2) “a” should be used instead of through”

A11. We have changed it to“However, most studies in the literature describe only a single family or case with exon 10 mutations”

- Materials and Methods:

Q12. “exon10” - missing space.

A12. We have added the space.

- Results:

Q13. The sentence: “In all, 74 of 133 participants who were found...” does not sound grammatically correct.

A13. We have corrected it to“Out of 133 participants, 74 cases carrying the RET germline mutations were diagnosed with MEN2A, including 26 males and 48 females.”

Q14. Consider breaking long multi-line sentences into shorter segments for better clarity of the text.

A14. We have rewrote the paragraph and broke long multi-line sentences into shorter segments to make the presentation clearer.

Q15. “pathologicaal” - extra “a” typo.

A15. We have deleted the extra“a”extra.

- References:

Q16. Reference numbers should have same font color.

A16. The font color of the references has been unified.

Q17. Adding “[J]” after journal title is not a part of a reference style used by PLOS

A17. All the“[J]”after journal title have been deleted.

Q18. The volume number should be separated from the year of publication by a semicolon, not a comma.

A18. The volume number and the year of publication has been separated from by a semicolon.

Q19. The references should include six author names before using et al., not three.

A19. We have added another three author names before using et al.

We appreciate for your warm work earnestly, and hope that the correction will meet with approval.

Once again, thank you very much for your comments and suggestions.

Yours sincerely,

Xiaoping Qi

---

## [Decision Letter · Decision Letter 1]

8 Jul 2025

Dear Dr. Qi,

Thank you for submitting your manuscript to PLOS ONE. After careful consideration, we feel that it has merit but does not fully meet PLOS ONE’s publication criteria as it currently stands. Therefore, we invite you to submit a revised version of the manuscript that addresses the points raised during the review process.

Your revised manuscript was reviewed by two experts in the field. Although one was satisfied with your revision, the other reviewer still identified many important problems in your submission. Please review the attached comments and provide thorough responses.

We look forward to receiving your revised manuscript.

Kind regards,

Yury E Khudyakov, PhD

Academic Editor

PLOS ONE

Reviewers' comments:

Reviewer's Responses to Questions

**Comments to the Author**

Reviewer #1: All comments have been addressed

Reviewer #2: All comments have been addressed

2. Is the manuscript technically sound, and do the data support the conclusions?

Reviewer #1: Yes

Reviewer #2: Yes

3. Has the statistical analysis been performed appropriately and rigorously?

Reviewer #1: Yes

Reviewer #2: Yes

4. Have the authors made all data underlying the findings in their manuscript fully available?

Reviewer #1: Yes

Reviewer #2: Yes

5. Is the manuscript presented in an intelligible fashion and written in standard English?

Reviewer #1: Yes

Reviewer #2: No

Reviewer #1: All recommendations from the reviewers were carefully considered and incorporated into the revised version. The manuscript demonstrates methodological soundness, clinical relevance, and scientific originality. Therefore, I recommend acceptance of the article for publication in its current form.

Reviewer #2: While the authors have done commendable work in addressing some of the specific points I raised earlier, the main issue of proper article presentation remains unresolved. I have identified multiple typos, grammatically incorrect and run-on sentences, and instances of awkward phrasing that limit the reader's ability to fully understand the article. I would emphasize again the point I have raised in the previous review: the authors should perform a comprehensive line-editing of the manuscript, as it still reads more like an advanced draft than a publication-ready paper.

The issues mentioned above include, but are not limited to, the following:

1) Typos — found in lines: 47, 75, 110, 128, 136, 227, 241, 276, and others.

2) Grammatical errors — in lines: 53, 90–92, 111, 178, 195–196, 210–212, among others.

3) Run-on and awkward sentences — in lines: 242–247, 265–270, 276–280, and more.

4) Figure legends — when embedded in the main text, they should be clearly separated to avoid confusion between the legend and the text itself.

5) There shouldn't be multiple empty lines between "References" title and previous section.

Additionally, I would like to point out a few technical issues in the article:

1) The "Methods: Detection of RET mutation" subsection would benefit from a brief description of the actual methods used, rather than referring readers only to previous articles. The methodology section should stand on its own, at least at a high level.

2) The authors should elaborate more about ref. 10, since they use its data extensively within the article. At a minimum, the Methods section should briefly explain what ref. 10 includes, why it was chosen for comparative analysis, and which factors could complicate direct comparisons between its data and the findings of this study.

3) The subsection "Results: Outcome of treatment" does not seem to include much information regarding connections between observed genotypes and treatment outcomes. While this section offers useful clinical information, the authors should stay focused on the core topic of the article — the association between *RET* mutations and oncogenic phenotypes.

4) The figures would benefit from more informative legends. Technical information, such as p-values of various comparisons, should be shown on the chart itself and does not need to be reiterated in the legend. Additionally, the figures would benefit from being in color rather than black and white, to improve readability.

5) Figure 2 — The y-axis should display the percent (or fraction) of pedigrees rather than absolute counts, since the compared groups have significantly different sample sizes.

6) SI: Medical Ethics Review Report - It is better to attach the translation as PDF, to ensure compatibility and ease of access across different devices.

**Do you want your identity to be public for this peer review?** For information about this choice, including consent withdrawal, please see our Privacy Policy

Reviewer #1: No

Reviewer #2: No

---

## [Author Response · Author response to Decision Letter 2]

19 Aug 2025

Dear Editor and Reviewers,

We are very grateful for your constructive and thoughtful comments on our manuscript “Genotype-phenotype correlations of germline mutations in exon 10 of RET proto-oncogene from 14 MEN2A families in Ethnic Han Chinese” (PONE-D-24-51352). Those comments are all valuable and very helpful for revising and improving our paper. We have studied comments carefully and have made correction which we hope meet with approval.

Response to Reviewer 2,

While the authors have done commendable work in addressing some of the specific points I raised earlier, the main issue of proper article presentation remains unresolved. I have identified multiple typos, grammatically incorrect and run-on sentences, and instances of awkward phrasing that limit the reader's ability to fully understand the article. I would emphasize again the point I have raised in the previous review: the authors should perform a comprehensive line-editing of the manuscript, as it still reads more like an advanced draft than a publication-ready paper.

The issues mentioned above include, but are not limited to, the following:

1) Typos — found in lines: 47, 75, 110, 128, 136, 227, 241, 276, and others.

2) Grammatical errors — in lines: 53, 90–92, 111, 178, 195–196, 210–212, among others.

3) Run-on and awkward sentences — in lines: 242–247, 265–270, 276–280, and more.

4) Figure legends — when embedded in the main text, they should be clearly separated to avoid confusion between the legend and the text itself.

5) There shouldn't be multiple empty lines between "References" title and previous section.

Answer to the

A. We have engaged a professional medical English editor to polish the language of the article and corrected all the linguistic errors you mentioned, and the proof is attached to the supporting informtion.

Additionally, I would like to point out a few technical issues in the article:

1) The "Methods: Detection of RET mutation" subsection would benefit from a brief description of the actual methods used, rather than referring readers only to previous articles. The methodology section should stand on its own, at least at a high level.

A1. Thanks for your suggestion. We have we have outlined the methodology. “Peripheral blood samples (5 mL) were collected from all 133 family members, including the probands, and then the Illumina HiSeq 2000 Analyzer was used for targeted sequencing. The targeted sequencing results were further validated by Sanger sequencing using an ABI 3700 Genetic Analyzer (Perkin-Elmer, Fremont, CA)”

2) The authors should elaborate more about ref. 10, since they use its data extensively within the article. At a minimum, the Methods section should briefly explain what ref. 10 includes, why it was chosen for comparative analysis, and which factors could complicate direct comparisons between its data and the findings of this study.

A2. Thanks for your suggestion. We have elaborated more about ref. 10 in the Methods section. “Data from the International RET Exon 10 Consortium, which included 340 patients from 103 families in 15 countries, were used for comparison with this study”

3) The subsection "Results: Outcome of treatment" does not seem to include much information regarding connections between observed genotypes and treatment outcomes. While this section offers useful clinical information, the authors should stay focused on the core topic of the article — the association between RET mutations and oncogenic phenotypes.

A3. Thanks for your suggestion. We have added more information between genotypes and treatment outcomes. “Post-operation, 48 of the 49 patients with MTC were confirmed histopathologically, while the remaining 1 patient (with the C618Y mutation) was confirmed as having CCH. All patients with the C609R, C618G/S/Y and C618S/R were confirmed to have positive lymph nodes. Whereas, the positive rate of lymph nodes in patients with the C611Y and C618R mutations was 7/14 (50%) and 12/16 (75%), respectively.”

4) The figures would benefit from more informative legends. Technical information, such as p-values of various comparisons, should be shown on the chart itself and does not need to be reiterated in the legend. Additionally, the figures would benefit from being in color rather than black and white, to improve readability.

A4. Thanks for your suggestion. We have moved the technical information from the legends to the charts. And the figures have been re-drawn in color.

5) Figure 2 — The y-axis should display the percent (or fraction) of pedigrees rather than absolute counts, since the compared groups have significantly different sample sizes.

A5. Thanks for your suggestion. The y-axis of Figure 2 has been displayed the percent.

6) SI: Medical Ethics Review Report - It is better to attach the translation as PDF, to ensure compatibility and ease of access across different devices.

A6. Thanks for your suggestion. The Medical Ethics Review Report has been attached to the SI as PDF.

We appreciate for your warm work earnestly, and hope that the correction will meet with approval.

Once again, thank you very much for your comments and suggestions.

Yours sincerely,

Xiaoping Qi,

18, Aug, 2025

---

## [Decision Letter · Decision Letter 2]

27 Aug 2025

Genotype-phenotype correlations of germline mutations in exon 10 of RET proto-oncogene from 14 MEN2A families in Ethnic Han Chinese

PONE-D-24-51352R2

Dear Dr. Qi,

We’re pleased to inform you that your manuscript has been judged scientifically suitable for publication and will be formally accepted for publication once it meets all outstanding technical requirements.

Kind regards,

Yury E Khudyakov, PhD

Academic Editor

PLOS ONE

Additional Editor Comments (optional):

Reviewers' comments:

Reviewer's Responses to Questions

**Comments to the Author**

Reviewer #1: All comments have been addressed

Reviewer #2: All comments have been addressed

2. Is the manuscript technically sound, and do the data support the conclusions?

Reviewer #1: (No Response)

Reviewer #2: Yes

3. Has the statistical analysis been performed appropriately and rigorously?

Reviewer #1: Yes

Reviewer #2: Yes

4. Have the authors made all data underlying the findings in their manuscript fully available?

Reviewer #1: Yes

Reviewer #2: Yes

5. Is the manuscript presented in an intelligible fashion and written in standard English?

Reviewer #1: Yes

Reviewer #2: Yes

Reviewer #1: The manuscript, in its R2 version, satisfactorily addresses all reviewer comments. There have been substantial improvements in clarity, methodology, and presentation of the results.

Reviewer #2: The authors made a great effort adressing the previous comments and bringing the article to the "publish-ready" form.

**Do you want your identity to be public for this peer review?** For information about this choice, including consent withdrawal, please see our Privacy Policy

Reviewer #1: No

Reviewer #2: No

---

## [Editor Report · Acceptance letter]

PONE-D-24-51352R2

PLOS ONE

Dear Dr. Qi,

I'm pleased to inform you that your manuscript has been deemed suitable for publication in PLOS ONE. Congratulations! Your manuscript is now being handed over to our production team.

Kind regards,

on behalf of

Dr. Yury E Khudyakov

Academic Editor

PLOS ONE